# PREHAB FAI- Prehabilitation for patients undergoing arthroscopic hip surgery for Femoroacetabular Impingement Syndrome -Protocol for an assessor blinded randomised controlled feasibility study

**Anuj Punnoose**[1,2]☯, **Leica Claydon-Mueller**[2]☯, **Alison Rushton**[3], **Vikas Khanduja**[4]*

**1** Young Adult Hip Service & Physiotherapy Department, Addenbrooke's- Cambridge University Hospitals NHS Foundation Trust, Cambridge, United Kingdom, **2** School of Allied Health, Anglia Ruskin University, Cambridge, United Kingdom, **3** Faculty of Health Sciences, School of Physical Therapy, Western University, London, Canada, **4** Department of Trauma and Orthopaedics, Young Adult Hip Service, Addenbrooke's – Cambridge University Hospital NHS Foundation Trust, Cambridge, United Kingdom

☯ These authors contributed equally to this work.

\* vk279@cam.ac.uk

**Data Availability Statement:** No datasets were generated or analysed during the current study. All

## Abstract

### Background

The past decade has seen an exponential growth of minimally invasive surgical procedures. Procedures such as hip arthroscopy have rapidly grown and become the standard of care for patients with Femoroacetabular Impingement Syndrome (FAIS). Although, the results of such procedures are encouraging, a large proportion of patients do not achieve optimal outcomes due to chronicity and deconditioning as a result of delay in diagnosis and increased waiting times amongst other factors. In a recent systematic review and meta-analysis of randomised control trials, moderate certainty evidence supported prehabilitation over standard care in optimising several domains including muscle strength, pain and health related quality of life in patients undergoing orthopaedic surgical interventions. However, the role of prehabilitation in patients with FAI syndrome undergoing hip arthroscopy has received little attention.

### Aim

To evaluate the feasibility, suitability, acceptability and safety of a prehabilitation programme for FAI to inform a future definitive randomised control trial to assess effectiveness.

### Methods

A systematically developed prehabilitation intervention based on a literature review and international consensus will be utilised in this study. A mixed methodology encompassing a two-arm randomised parallel study alongside an embedded qualitative component will be used to answer the study objectives. Patients will be recruited from a tertiary referral NHS

relevant data from this study will be made available upon study completion.

**Funding:** Mr Anuj Punnoose has received funding for his PhD Fellowship from the National Institute of Health & Care Research UK (Award NIHR302117). The funders did not and will not have a role in study design, data collection and analysis, decision to publish, or preparation of the manuscript.

**Competing interests:** I have read the journal's policy and the authors of this manuscript have the following competing interests: Dr Khanduja reported being a chief investigator for the NIHR doctoral fellowship during the conduct of the study; receiving consulting fees from Smith & Nephew PLC and Arthrex Inc, outside the submitted work; serving as a board member for the UK Non-Arthroplasty Hip Registry, British Hip Society, European Society of Sports Traumatology, Knee Surgery, and Arthroscopy, and the International Society of Orthopaedic Surgery and Traumatology; and serving on the editorial boards of the Journal of Clinical Orthopaedics and Trauma, International Journal of Orthopaedics, International Orthopaedics, and Open Access Journal of Sports Medicine. This does not alter our adherence to PLOS ONE policies on sharing data and materials. No other disclosures were reported.

centre for young adult hip pathology in the UK. Patient reported outcomes such as iHOT-12, Brief Pain Inventory Scale (Short form), Hospital Anxiety and Depression Scale and Patient Global Impression of Change score will be obtained alongside objective measurements such as Muscle Strength and Star Excursion Balance Test at various time points. Outcome measures will be obtained at baseline (prior to prehabilitation intervention), after prehabilitation before surgery, and at 6 weeks+/- 4 weeks and 6 months +/- 4 weeks (planned primary endpoint for definitive RCT) postoperatively when participants attend the research site for clinical care and remotely at 12 months +/- 4 weeks postoperatively. Mean change and 95% CI, and effect size of outcome measures will be used to determine the sample size for a future RCT. For the qualitative component, in depth face-to-face semi-structured interviews with physiotherapists and focus groups with participants will be conducted to assess the feasibility, suitability, and acceptability of the prehabilitation intervention using a predetermined success criteria. All qualitative data will be recorded, transcribed verbatim and thematically analysed.

## Discussion

This study will be first of its kind to evaluate a systematically developed prehabilitation intervention for patients with FAIS undergoing hip arthroscopy. This study will provide important preliminary data to inform feasibility of a definitive RCT in the future to evaluate effectiveness of a prehabilitation intervention.

## Trial registration

ISRCTN 15371248, 09/03/2023.

## Trial protocol

Version 2.3, 26th June 2023.

## Introduction

Femoroacetabular Impingement syndrome (FAIS) is a well-recognised pathological entity and a common cause of hip and groin pain in young adults [1]. The pathology encompasses a morphological abnormality of the femur (Cam) or the acetabulum (Pincer) and can lead to chondro-labral dysfunction and eventually in some cases early onset of osteoarthritis [2]. An epidemiological study on 200 patients aged 16–65 years, representative of the UK population, reported a high prevalence of FAI with 47% (95% CI [42–51]) showing signs of cam morphology [3]. Additionally, deficits in muscle strength and range of motion (ROM) may lead to altered or increased hip joint loads causing detrimental effects [4, 5]. Non-surgical management such as physiotherapy could assist in modifying hip joint loads and address the above physical impairments, although, evidence of the effectiveness of such interventions are currently lacking [6, 7].

Excision of the impingement lesion (Cam, pincer or mixed) and addressing chondro-labral pathology through surgical procedures like hip arthroscopy may be necessary to provide better medium to long term outcomes [8]. Innovations in diagnosis and surgical technique have resulted in an exponential growth of hip arthroscopy procedures in the past decade [9]. Furthermore, two recent multicentre randomised controlled trials (RCTs) have found significant

improvement in outcomes relating to daily activities and quality of life at 8–12 months follow-ing hip arthroscopy compared to physiotherapy [10, 11]. The mean between group difference in the iHOT-33 scores was 6·8 [95% CI: 1·7–12·0, p = 0.0093] in favour of hip arthroscopy which exceeded the minimum clinically important difference (MCID) of 6.1 [10]. However, results from these trials demonstrate that only half (51%) of participants in the hip arthroscopy group showed significant improvement on the primary outcome measure–iHOT33 which relates to quality of life at 12 months post-op [10]. Another study from the Non Arthroplasty Hip Registry in the UK of over 4900 patients with FAI, showed only 35% of all patients improve or achieve optimal outcome post-operatively [8]. These findings may suggest that fac-tors other than surgery may contribute to their post-operative outcomes.

Several factors have been shown to influence the outcomes after hip arthroscopy. These include, the presence of OA, severe dysplasia, inadequate removal of the impingement lesion, and soft tissue injury sustained during the procedure [2, 12, 13]. Amongst the non-surgical fac-tors are prolonged waiting times leading to the chronicity and deconditioning of the muscles [14], deficits in muscle strength [15], psychological distress [16] and presence of concomitant pathologies like gluteal tendinopathy [17] and athletic pubalgia [18]. Therefore, early identifi-cation and optimisation prior to surgery will be crucial in delivering optimal post-operative outcomes.

Pre-operative rehabilitation intervention or 'prehabilitation' is the process of enhancing a patient's functional capacity prior to surgery in order to improve post-operative outcomes [19]. Prehabilitation interventions have been tested and found to be beneficial across all age groups and various surgical pathways including orthopaedics [20, 21]. It is important for inter-ventions included in the prehabilitation programme to be specific to the clinical condition and tailored to patient needs [20]. A person-centred approach, which allows patients to take con-trol of their own outcomes through prehabilitation, places the patient at the core of their peri-operative pathway. This will heighten their motivation to make positive behavioural changes during the pre-operative phase and provide pre, peri and post-operative benefits [22].

Although the effectiveness of prehabilitation has been investigated in a wide variety of orthopaedic conditions, its effectiveness in patients undergoing hip arthroscopy has received little attention. Recently, a small pilot study on patients with FAI (n = 18) favoured prehabilita-tion over standard care [23]. However, it is not known if benefits are maintained over a longer period and the study demonstrated a high risk of bias (randomisation process, concealment method, baseline differences between groups). Additionally, although participants were recruited from the NHS, the programme was delivered in a private setting and interventions were not developed using a scientifically robust methodology. The study also did not report on adherence to the exercise interventions, an important issue as low levels of adherence can limit the effectiveness of exercise [24]. These findings suggests that there is a need for a well devel-oped study to assess feasibility, suitability, acceptability and safety of a prehabilitation interven-tion within the NHS setting prior to a definitive trial to evaluate effectiveness.

## Aim

To evaluate the feasibility, suitability, acceptability and safety of a prehabilitation programme for patients with FAIS to inform a future definitive randomised controlled trial to assess effectiveness.

## Objectives

The objectives of the study and success criteria are detailed in Table 1.

**Table 1. Study objectives and success criteria.**

| General objectives | Success criteria |
|---|---|
| **Recruitment procedure** | Participants were recruited within the time constraints of the local NHS hospital |
| | Participants report that there were no challenges with the recruitment procedure |
| **Data collection methods** | Data completeness of ≥80% |
| | Patients and assessors reported that there were no challenges with the data collection methods |
| **Follow-up procedures** | 100% of participants were contacted for follow-up |
| | ≥80% completion of follow-up outcome measures |
| **Specific objectives** | **Success criteria** |
| **Feasibility** | |
| **Participant recruitment rates** | The recruitment rate of this study will be considered sufficient if at least 33% of eligible patients are recruited over a period of 18 months. |
| **Attrition rate** | <30% dropout |
| **Usability of Physitrack** | Patients and Physiotherapists reported no challenges with the use of Physitrack and telehealth system. |
| **Capacity (time and effort) of clinicians to deliver the programme** | Physiotherapists report that they had adequate time and resources available to deliver the programme |
| **Treatment fidelity** | ≥80% completion of the self-reported checklist |
| | Physiotherapists reports that there were no barriers identified in the use of self-reported checklists. |
| **Suitability** | |
| **Outcome measures** | Patients and assessor report that the outcome measures were appropriate and self-explanatory |
| | Data completeness of ≥80% |
| **Adherence to the programme** | ≥70% of the sessions (face to face + home ex monitored by Physitrack) |
| **Time required to undertake each stages of the study** | Physiotherapists report that they had enough time to complete each stages of the study. |
| **Service infrastructure** | Recruitment targets met<br>Data completeness of ≥80%<br>Clinicians and researchers report that there was adequate infrastructure to allow completion of a full trial. |
| **Acceptability** | |
| **Intervention** | Patients and clinicians report that the programme was appropriate and satisfactory |
| **Randomisation** | Patients are willing to be randomised for a future definitive trial. (≥80%) |
| **Safety** | |
| **Intervention** | Patients report that the intervention was safe, and no serious adverse events occurred during the prehabilitation phase. |

## General objectives

To assess the feasibility, suitability, acceptability and safety of a prehabilitation intervention evaluating:

- Recruitment procedures

- Data collection methods

- Follow-up procedures

- Determine sample size for a full trial to test its effectiveness

## Specific objectives

### Feasibility.

- To evaluate recruitment procedures such as participant recruitment and attrition rates [25]

- To evaluate follow-up rates and response rates to questionnaires [25]

- To evaluate the ease of using the Physitrack app [26]

- To evaluate the time and effort required by clinicians to deliver the intervention [25]

- To evaluate fidelity of intervention delivery [27]

### Suitability.

- To evaluate suitability of the outcome measures [28]

- To evaluate participant adherence with prehabilitation interventions [25]

- To evaluate the time taken to undertake each phases of the study [25]

- To evaluate the appropriateness of incorporating a prehabilitation intervention for FAI into the current NHS services infrastructure [29]

### Acceptability.

- To evaluate the acceptability of interventions to patients and clinicians [28]

- To evaluate the willingness of patients to participate in a full RCT in the future [28]

### Safety.

- To evaluate the safety of the intervention [25]

## Methods

This feasibility study protocol will be reported as per the CONSORT 2010 statement: extension to randomised pilot and feasibility trials to ensure transparency and reproducibility [30] and has been registered on the ISRCTN database. This study has received ethics approval from East of England—Cambridge Central Research Ethics Committee (IRAS ID: 293927, REC Ref 23/EE/0024 dated 01/02/2023) and began recruitment from the 21st of April 2023. Further amendments (SA 01, dated 10/2/2023; SA 02, dated 20/04/23, SA 03, dated 28/06/23) to the protocol are specified below. The Sponsor and Ethics committee will be informed of any future changes to the protocol and the Trial registry will be updated accordingly.

## Protocol amendments

1. Study follow-up points will be +/- 4 weeks to accommodate flexibility in clinic appointment within the NHS. Upon further consideration of the current situation and waiting times within the NHS, having a rigid follow-up point would not be feasible. Therefore, the follow-

up points are kept flexible, but within a reasonable duration, so this does not affect the quality of the study.

2. Patient reported outcome measures will be completed digitally via online surverys at baseline and all other follow-up points (at pre-op, 6 weeks,6 months and 12 months post-op) using REDCap. Participants will receive an email containing a link to the online outcome measure survey along with instructions for completion during the specified follow-up intervals mentioned above. If participants forget to complete the outcome questionnaire on the required day, a reminder to complete it will be sent on two more occasions after the deadline to facilitate compliance.

3. The REDCap database will be used to collect and store the data for the study and will therefore be more secure and will strictly follow all UK and GDPR guidelines.

4. Both PI and Chief Investigator are part of the direct clinical team and routinely collect and review data regarding patient demographics, clinical examination details and details of any investigations done for their care. The above details will be collected and utilised for the purpose of the study. This information is mentioned on the current PIS, however not explicitly and covering all the above terms mentioned. This will be stored securely in the hospital computers and REDCap database as mentioned above. If a need arises to share data or present at meetings for dissemination, this data will be anonymised, and no patient identifiable details will be presented.

5. Following baseline evaluation, participants will be randomly allocated to one of the two groups (prehabilitation or usual care) by the PI using a computer-generated simple random allocation sequence. This will be uploaded to REDCap and all study personnel will be blinded to the allocation sequence ensuring adequate concealment.

6. Pre-op education session will be delivered virtually via Zoom/MS Teams. This is to reduce the impact on patient travel and reduce the need for additional resources e.g room availability in the hospital

7. Two or more focus group (8–10 participants) will be conducted as needed. This is to improve data quality and may achieve data saturation in a robust manner improving the overall quality of the study. This will only be implemented with participant's explicit consent.

## Design

A randomised controlled study design (RCT) is considered as gold standard when evaluating the effectiveness of an intervention as it eliminates various biases prevalent in other study designs [31]. However, a RCT can be challenging with several unknown factors at play (recruitment, adherence, attrition, acceptability and suitability of interventions). Therefore, prior to evaluating the effectiveness of an intervention, researchers are encouraged to assess its feasibility, suitability, acceptability, and safety, and optimise the overall design of the trial to ensure any uncertainties are addressed prior to a full trial [27, 32].

This feasibility trial will use a mixed methodology encompassing the following;

- A quantitative two arm parallel group feasibility trial

- An embedded qualitative component consisting of semi-structured interviews with physiotherapists and patient focus groups using thematic analysis

## Quantitative component of study

It is imperative that steps are undertaken to improve the methodological rigour right at the initial stages of any research study [33]. Consistent with the Medical Research Council Framework recommendations for developing and evaluating complex interventions [32], it is crucial to develop the intervention prior to testing its efficacy or effectiveness. A recent systematic review and meta-analysis conducted by the lead authors Punnoose & Claydon-Mueller et al., analysed 48 unique trials investigating the effectiveness of prehabilitation in patients undergoing orthopaedic surgery [21]. This study included both published and unpublished trials until June 2022 and synthesised evidence on the effectiveness of prehabilitation on several outcome domains including quality of life, pain, muscle strength and function, laying a thorough foundation to the prehabilitation intervention used in this study. Alongside underscoring the effectiveness of prehabilitation, the study also reported on the optimum duration, dosage and frequency of prehabilitation, and emphasised on the importance of reporting adherence to the exercise intervention. However, majority of studies (85%, n = 41) included in the systematic review were on joint replacements, highlighting the lack of good quality research around prehabilitation effectiveness in other orthopaedic surgical procedures such as Anterior cruciate ligament reconstruction and FAIS.

Due to the lacunae of literature around prehabilitation components and its parameters within the FAIS population, an expert panel consensus statement with 17 international experts defined the core components of the intervention [34]. The study recommended 6 core domains to be targeted during the pre-operative phase- muscle strength, range of motion, proprioception,cardiovascular fitness, anxiety and depression and addressing any co-existing pathologies alongside FAIS. Additionally, the consensus statement recommended that prehabilitation for FAIS be delivered for 6–8 weeks as a combination of face to face and virtually via telehealth. This study is currently in its preparatory stages of publication. The resulting prehabilitation intervention developed though this rigorous process are detailed in S1 Appendix.

A prospective mixed methods, two group parallel feasibility study will be conducted to answer the above objectives [28]. Fig 1 shows the schedule of enrolment, interventions, and assessments for the study. Objective outcome measures will be obtained at baseline (prior to prehabilitation intervention), after prehabilitation before surgery, and at 6 weeks+/- 4 weeks and 6 months+/- 4 weeks (planned primary endpoint for definitive RCT) postoperatively when participants attend the research site for clinical care. Additionally, all patient reported outcome measures will be captured at 12 months+/-4 weeks post-op. Physical outcome measures will be assessed onsite during the participant's clinical visits and patient reported outcomes will be collected digitally via REDCap (Research Electronic Data Capture) [35, 36] (see Fig 2). The online outcome measure survey will be built using REDCap enabling real time data to be captured and stored securely. Participants will receive an email containing a link to the online outcome measure survey along with instructions for completion during the specified follow-up intervals mentioned above. If incomplete, two additional reminders will be sent to the participants after the deadline to facilitate compliance.

**Participants.** The study will be conducted at a single tertiary Young Adult Hip Service at a National Health Service (NHS) hospital. Potential participants will be identified from the waiting list for hip arthroscopy at the NHS site. This will be performed by the Chief Investigator or the Principal Investigator (PI) who is a part of the direct clinical care team. Potential participants for the study will be contacted if their surgical date in the future will allow appropriate time for the consent process, randomisation and 8 weeks of prehabilitation intervention.

**Table 2. Schedule of enrolment, interventions, and assessments**

| TIMEPOINT | Enrolment Pre-randomisation | Allocation Randomisation | Post-allocation Outcome measures | Treatment delivery | Qualitative component | Pre-op (after prehab) | 6 weeks post-op | 6 months post-op | 12 months post-op | Close-out 12 months post-op |
|---|---|---|---|---|---|---|---|---|---|---|
| **ENROLMENT:** | | | | | | | | | | |
| Eligibility screen | x | | | | | | | | | |
| Informed consent | x | | | | | | | | | |
| Baseline demographic data | x | | | | | | | | | |
| Allocation | | x | | | | | | | | |
| **INTERVENTIONS:** | | | | | | | | | | |
| Prehabilitation + education | | | | ←——→ | | | | | | |
| Usual care- No prehabilitation | | | | N/A | | | | | | |
| **ASSESSMENTS:** | | | | | | | | | | |
| iHOT-12 | | | x | | | x | x | x | x | x |
| BPI (Short Form) | | | x | | | x | x | x | x | x |
| HADS | | | x | | | x | x | x | x | x |
| PGIC scale | | | x | | | x | x | x | x | x |
| Hip muscle strength | | | x | | | x | x | x | | |
| Trunk endurance | | | x | | | x | x | x | | |
| SEBT | | | x | | | x | x | x | | |
| Feasibility measures | | | | | | | | x | | |
| Adverse events | | | | | | x | x | | | |
| Focus group with research participants | | | | | x | | | | | |
| Semi-structured interviews with Physiotherapists | | | | | x | | | | | |

BPI- Brief Pain Inventory, HADS- Hospital anxiety and depression scale, iHOT- International Hip Outcome tool, N/A- Not applicable, PGIC- Patient Global Impression of change, SEBT- Star excursion Balance Test

**Fig 1. Schedule of enrolment, interventions, and assessments.**

A Patient Information Sheet (PIS) will be provided to participants to facilitate the consenting process. The research team will ensure that they adequately explain the aim, study treatment, potential benefits or harm of taking part in the trial to the participants. They will also re-iterate that participation is voluntary and participants can withdraw from the study at any time, without giving a reason. Participants will be phoned after 1 week from when they have received the PIS to see if they are interested in taking part. For patients interested in participating, written consent to participate in the study will be confirmed at first face to face contact during their routine pre-op assessment. If this is not possible due to any reason, the consent

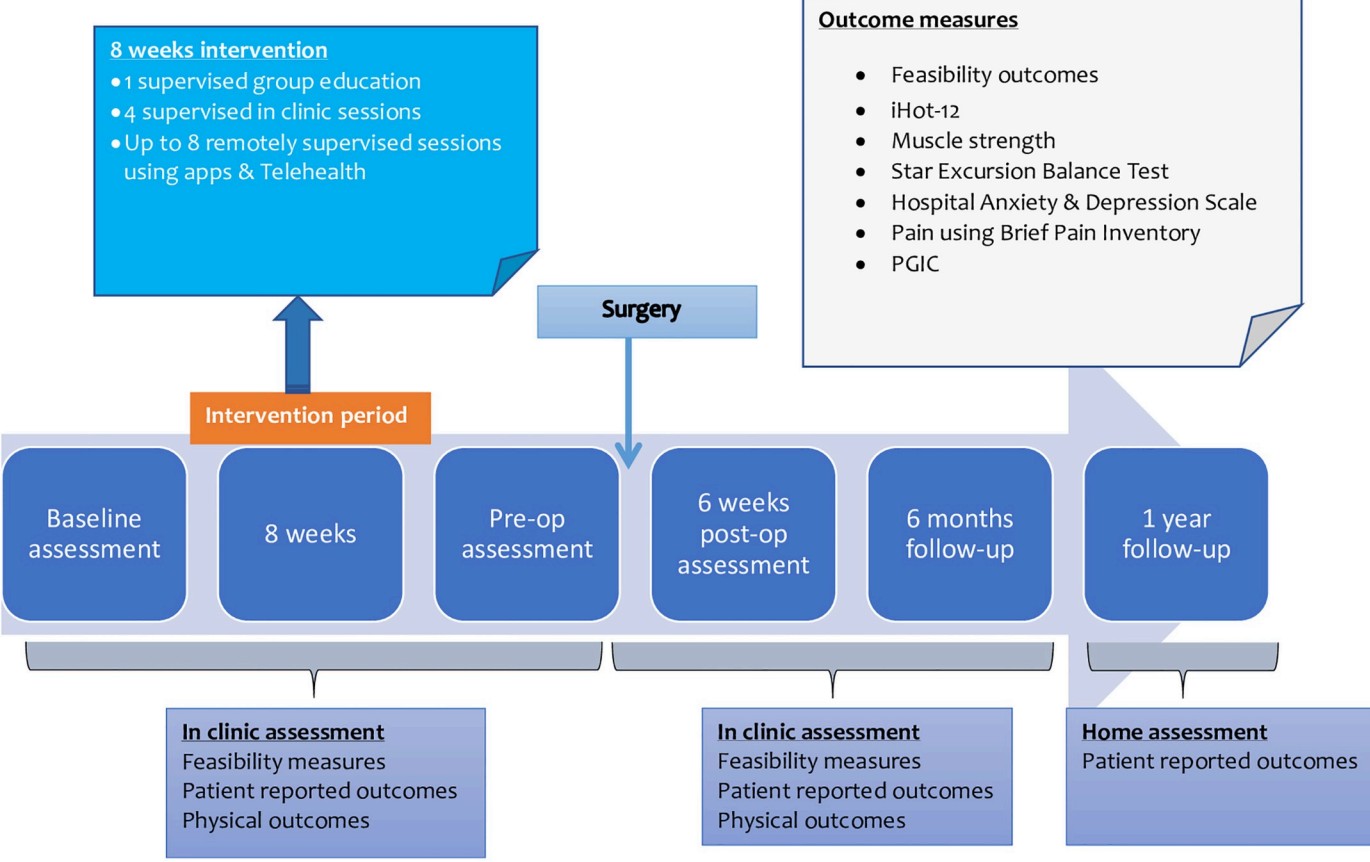

**Fig 2. Timeline for interventions and outcome measures.**

process will be completed over the phone. The consent process will be witnessed by a member of staff independent of the trial team. The PI (or delegate) will sign the consent form as the person receiving consent and the independent witness will sign the consent form to confirm that the consent process was followed and the participant gave their verbal consent to take part. Participants will be asked to sign the consent form when they come for their surgery. A copy of the consent form will be given to them at this point. Participants will be provided with the current version of the REC approved patient information sheet to read. Participants will have the opportunity to pose any questions, and consent will be obtained by the Principal Investigator or a qualified member of the clinical care team only after addressing all of their inquiries. Participants can withdraw their consent at any point during the study. This will be recorded in their electronic patient records. A sample of the consent form can be found in S2 Appendix

### Eligibility criteria

**Inclusion criteria.** All patients referred to Cambridge Young Adult Hip Service (aged 16–50 years), undergoing hip arthroscopy for the management of FAIS.

**Exclusion criteria.**

- Previous hip disease such as Perthe's, Slipped upper femoral epiphysis or avascular necrosis [10]

- Participants who are unable to give full written consent

- Participants who are not fluent in English

- Pre-existing neuromuscular conditions like Motor Neuron Disease or Multiple Sclerosis

- History of any previous hip surgery

- History of any previous hip arthroscopies

## Interventions

All participants, regardless of group allocation, will receive standard peri & post-operative care and rehabilitation. A record of their analgesia consumption will be noted during their follow-up appointments (pre-op, post-op 6 weeks and 6 months) in a logbook built in REDCap by a member of the research team.

**Prehabilitation intervention.** Interventions are described based on the Template for Intervention Description and Replication (TIDieR) to allow reproducibility [37]. The intervention will be delivered over a period of 8 weeks prior to surgery and will target 5 domains-muscle strength, range of motion, proprioception, cardiovascular fitness and addressing co-existing pathologies. Literature suggests high prevalence and poorer outcomes of depression and anxiety in people with FAI [16, 38], and the intervention will therefore also include an educational session delivered virtually via Microsoft Teams software by the PI to provide greater understanding of the surgery to alleviate anxiety and help manage patient expectations. The intervention will include at least four in-clinic face-to-face sessions (once a fortnight) with an experienced physiotherapist (>2 years' experience in treating musculoskeletal conditions including FAIS) followed by upto six to eight remotely monitored sessions (once a week) using a telehealth system (Fig 2). The number of sessions will be determined based on physiotherapist assessment and intensity and complexity of exercises will be gradually increased depending on each participant's individual progress. Due to the wide variation of patient characteristics within the FAIS population (e.g. sedentary or athletic) interventions will be tailored according to individual participant's capabilities assessed by their treating physiotherapist. Components of the prehabilitation interventions are designed not to aggravate patient symptoms. However, in case of worsening of symptoms, the physiotherapist will re-evaluate and adapt the exercises to ensure safety. Details of the prehabilitation intervention is described in S1 Appendix.

Physitrack provides exercise information and videos via website and apps. Dosage and frequency of the exercises can be selected, and patients will be encouraged to record completion of the exercises enabling the physiotherapist to measure adherence via the app. Physitrack has been found to improve adherence and patient confidence and is utilised across several NHS Trusts in the UK [39]. Additionally, the use of Physitrack's tele-rehabilitation would enable physiotherapists and patients to interact with each other via video providing reassurance of the correct exercise techniques and progression [40].

The participants in the intervention group will use Physitrack which provides photos and videos of the prescribed exercises and those unable to use the app will be provided with paper-based instructions. Additionally, patients will be asked to complete their exercises at home twice weekly.

The prehabilitation intervention will be delivered by experienced physiotherapists as described above. All the rehabilitation components are standard clinical therapies familiar to a trained physiotherapist. The physiotherapists will undergo training (by PI) on how to deliver these components together as per the protocol for this study. A representative from Physitrack will be invited to deliver training on the use of the app.

**Usual care intervention.** As per the host hospital's current clinical guidelines, the usual care intervention consists of advice to continue with normal activities and no prehabilitation therapy.

## Randomisation and blinding

A simple random allocation sequence will be generated using GraphPad QuickCalcs website [41] and uploaded to REDCap. Once patients are allocated to one of the two groups (prehabilitation or standard care), all baseline data will be obtained. All study personnel will be blinded to the allocation sequence ensuring adequate concealment [35, 36].

Blinding participants and clinicians delivering interventions will not be possible as both will be aware of the intervention and allocation. However, the content of the programme will only be known to participants in the intervention group thereby reducing the risk of cross contamination between groups [42]. Physical outcome measures will be measured by a blinded assessor who will be trained by the PI to ensure standardisation and improve reliability of the assessment.

## Outcome measures

All physical and patient reported outcome measures will be collected at various time points as explained above and in Figs 1 and 2.

**Primary clinical outcome measure.** *iHOT-12.* The iHOT-12 is a patient reported outcome of Quality of Life (QoL) [43] and will be the planned primary outcome measure for this study. This shorter version of the original iHOT-33 was developed and validated for patients with FAIS and was found to be valid, responsive and a reliable tool to measure the impact of hip disease in young active individuals on QoL [43].The minimal clinical important difference for the iHOT-12 is 13.0 and patient acceptable symptom state threshold is 63.0 for FAIS [44].

**Secondary clinical outcome measures.** *Muscle strength using handheld dynamometer.* Several studies have discussed the implications of reduced muscle strength in the hip and how this could predispose to lower extremity injuries [45–47]. A recent systematic review evaluating muscle strength measurement in various hip pathologies reported that there is evidence of bilateral hip muscle weakness regardless of whether the hip joint is symptomatic or not [48]. Both motor driven and hand-held dynamometry (HHD) are reliable methods of muscle strength measurement and should be used with make tests (where the patient pushes against the examiner's fixed resistance) [48]. For ease of use in a clincal setting, a HHD will be used in this study. HHD with an external belt fixation has shown good inter-tester reliability with intra-class coefficient of 2.1 (range 0.76 to 0.95) [49]. Maximum voluntary isometric contraction will be utilised as the measurement for all hip muscles groups [50]. A minimum detectable change (MDC) score is often used to detect real change in an outcome measure overtime [51]. An MDC value at 95% CI ranging from 9–45 points was reported for individual hip muscle groups in healthy subjects using a HHD showing it is responsiveness [52]. Further details on how to perform the tests are described in S3 Appendix.

*Star Excursion Balance Test (SEBT).* The SEBT is a reliable measure to assess dynamic postural control and proprioception and has shown excellent inter-rater reliability [53]. The SEBT has shown good criterion and divergent validity in relation to pain, hip strength and ROM in patients with FAI [54, 55]. See S3 Appendix.

*Brief Pain Inventory-short form (BPI).* Most patients with FAI experience pain in the hip or groin [56], often chronic in nature. BPI is a multidimensional scale that can reliably measure chronic pain and its interference with an individual's physical and social functioning [57]. The tool is responsive to change in pain associated with pharmacological, physical as well as

psychological interventions and was recommended by the Initiative on Methods, Measurement, and Pain Assessment in Clinical Trials [58].

*Hospital Anxiety and Depression Scale (HADS).* Recent studies have shown the prevalence of anxiety and depression in patients undergoing hip arthroscopic procedures [16, 38]. Previous studies have reported improvements in the level of anxiety and depression experienced by patients when undergoing prehabilitation [59, 60]. The HADS scale is a validated tool to measure anxiety and depression in general medical population of patients and will be utilised in the study [61]. A literature review including 747 papers reported a sensitivity and specificity of 0.8 at a cut off score of >8 on HADS which was very similar to other health questionnaires [62].

*Patient's Global Impression of change (PGIC).* The Global rating of change (GRC) scales are often used in research particularly within the musculoskeletal area [63]. Among the GRC scales, PGIC scale is frequently used to gather a patient's perception of change after an intervention. The scale typically consists of 7 points which depicts a patient's overall improvement from "very much improved" to "very much worse [64]. These scales have shown adequate reproducibility and sensitivity to change in a variety of disorders, and are easy to use and interpret [63].

## Participant demographic data

Participant baseline demographic data including age, gender, height, weight, Body Mass Index, duration of symptoms and work status will be recorded on REDCap once they have consented to participate in the study. Additionally, details of their physical examination, investigations and past medical history recorded as a part of their routine care will be collected and used for the purpose of this study.

## Sample size

An overall sample size of 24–50 is recommended to estimate the standard deviations for calculating sample size required for a full trial [65–67]. As this is a feasibility study, the sample size will be based on recruitment rate over a period of 18 months. As a tertiary referral centre for hip arthroscopy, we anticipate around 48 eligible patients over a period of 18 months. We could estimate a recruitment rate of 33% (i.e. 16 participants) to take part in the study within a 95% confidence interval of ±10.5%.

## Data analysis

A CONSORT diagram will be used to describe and analyse recruitment and attrition rates [30]. Data will be analysed using SPSS version 28. Between group baseline demographic data will be analysed using independent t-tests for continuous variables and chi-square tests for categorical variables. Descriptive statistics, such as percentages (for rate calculations), means, standard deviations and mean change (if data are normally distributed) will be analysed to evaluate distribution of scores and analyse the floor and ceiling effect of the primary outcome measure. Floor and ceiling effects are defined the percentage of participants who scored the minimum (floor) and maximum (ceiling) possible score in an outcome measure [68].

Data will be reported using Intention to treat analysis [69]. Standard deviations and effect sizes for the primary outcome measure will be used to calculate the required sample size for a definitive trial in the future.

This is a feasibility trial of exercise interventions and will carry minimal risks. Therefore, a formal data monitoring committee is not required.

## Qualitative component of study

### Design

An embedded qualitative component will be utilised to answer specific trial objectives and to refine and adapt the design prior to the full RCT. Prior to participating in the qualitative study, participants will be provided with an information leaflet and opportunity will be given to raise any questions to the researchers regarding the processes. Consent from all participants will be obtained prior to taking part in the interviews and focus groups.

### Physiotherapist participants

In-depth face to face semi structured interviews will be used to explore the views of the physiotherapists (n = 4) regarding the feasibility, suitability and acceptability of the intervention, outcome measures as well as the Physitrack app in an NHS setting as described below. Physiotherapists involved in the delivery of the interventions and collection of outcome measures will be invited for this purpose. The interviews will be conducted by the PI within one year of commencement of the study. Appropriate questions for the interviews will be developed by the PI and the supervisory team prior to conducting the interviews. A patient and public involvement group will review the questions for clarity and appropriateness [70].

### Research participants

Two or more focus groups with research participants will be conducted until data saturation has been achieved following the 6 months assessment point to evaluate the research objectives. A purposive sample of 8–10 participants based on characteristics such as group allocation, adherence and time in their clinical pathway (pre-op & post-op) will be included as recommended in the literature [71]. A predetermined topic guide developed by the PI and reviewed by the PPI panel will be used for the focus group. The focus groups will be conducted by 2 researchers: the PI (facilitator) and a supervisor (observer).

### Data analysis

All qualitative data will be recorded and transcribed verbatim. The personal assistant to the Chief investigator (VK) will be responsible for the transcriptions. As a member of staff, they will follow all guidelines of confidentiality and information governance procedures as per the host hospital's policy. Once used, all original recordings will be deleted. Quality of the data will be ensured via practitioner and assessor training, data completeness and accuracy.

All transcribed qualitative data will be thematically analysed [72]. It is anticipated that the qualitative data captured will have a diverse range of opinions due to the novel nature of the intervention and an inductive approach will be best suited for this purpose [72]. Transcripts will be reviewed by the participants for credibility [73] prior to analysis by the researchers using Nvivo 14 software [74]. For credibility and confirmability of the qualitative data, coding and themes will be analysed by the PI and re-checked by the primary academic supervisor (LCM).

## Feasibility, suitability, acceptability and safety of the intervention

A pre-determined success criteria will be used to analyse the quantitate and qualitative data (Table 1). The study objectives will be considered successful if the success criteria were satisfied upon integration of data derived from both quantitative and qualitative methods.

## Feasibility measures

Feasibility parameters will be utilised to answer the objectives as per Table 1. The recruitment rate will be determined by the number of participants who are eligible and consent to participate in the study. Attrition will be defined by the number of consenting participants who drop out during the study. Timing of dropouts and reasons (where provided) will be explored to determine if the follow-up points are appropriate prior to a definitive RCT. Follow-up rates and response rates to questionnaires will be collected at each follow-up point. The usability of the Physitrack app and tele-rehabilitation will be evaluated using a modified System Usability Scale (SUS) [26]. Time and effort of clinicians delivering the interventions will be captured via the hospital's electronic system.

Treatment fidelity is the extent to which an intervention is delivered as per study protocol and is critical in the development and testing of evidence-based interventions [75]. Assessing fidelity in the feasibility stage is also critical to identify interventions that have lower than expected fidelity which can then be used to refine interventions prior to a full trial [76]. Fidelity will be monitored by analysing self-reported checklists completed by the treating Physiotherapists. A numerical rating scale (5 point Likert scale) will be used to measure content and quality of the interventions and post-session feedback will be given to the clinicians to improve treatment delivery [77].

## Suitability measures

Physiotherapists and participants will be asked the appropriateness of outcome measures as well as time and effort required to complete them. Adherence to the intervention will be measured by capturing the number of exercises sessions delivered face-to-face and home exercises remotely monitored by Physitrack app. Qualitative data from the interviews and the focus group will also be collected to develop a training manual and to establish if there is adequate resources and infrastructure at the host hospital to support a definitive trial in the future.

## Acceptability measures

Acceptability of the interventions will be evaluated during focus groups with participants using qualitative methods.

## Safety measures

Safety of the interventions will be determined based on the number and nature of adverse events. All adverse events (related or unrelated to interventions) will be recorded. Components of the prehabilitation interventions are designed not to aggravate patient symptoms. However, in case of worsening of symptoms, the physiotherapist will re-evaluate and adapt the exercises to ensure safety.

## Data storage

Data collected will be stored safely on REDCap database which can be accessed only by the study investigators at the host institution. Additionally, personal data of participants will be stored securely on EDGE- a clinical research management platform approved by the host institution and will follow GDPR (General Data Protection Regulations) guidelines [78]. Monitoring and auditing of the data by the sponsor hospital's R&D might be required during the study, however, all patent identifiable data will be removed prior to sharing the data. All personal data used for the study will be destroyed at the end of the study. Pseudo-anonymised quotations from the respondents might be used for dissemination such as peer reviewed

publications or conferences. No single patient data will be reported or published ensuring confidentiality.

## Data availability

No datasets were generated or analysed during the current study. All relevant data from this study will be made available upon study completion.

## End of the study

The end of the study will be declared when the last participant in the study has completed their 1-year post-op follow-up.

## Ethics and dissemination

This study raises no major ethical or legal issues. The randomised study design will provide equal opportunities for all participants to treatment allocation. Each participant in the study will be given an opportunity to read and understand the study prior to signing the consent form and they will have the right to withdraw from the study at any time. The intervention will be delivered as a combination of face-to-face and remotely using an app giving equal opportunities for patients who are unable to travel often. Reimbursements for their travel has been included and approved by the funding organisation.

Additionally, participation in the study will not alter their clinical pathway. Outcome measures will be taken at the time of their normal clinical visits reducing their visits to the hospital.

The results of this study will be disseminated to improve awareness and engagement of all parties (patients, research community and the public) in preparation for the future trial. The study results will be shared with the research and scientific community via peer-reviewed publications, open access repositories and scientific meetings and conferences.

## Patient and Public Involvement (PPI)

A focus group involving a patient and public panel informed the design of the study. An advisory group will be formed with two people from the PPI panel who will oversee the study. Several activities are planned with the PPI panel during the study as below;

1. Co-design the prehabilitation leaflet

2. Develop question guides for the research focus groups and semi structured interviews

3. Co-develop a training manual for clinicians

4. Design dissemination materials

All patient and public involvement meetings will be evaluated using an Impact log and reported in line with the GRIPP2 short form reporting checklist [79].

## Strengths of the study

To our knowledge, this is the first study to test the feasibility, suitability, acceptability and safety of a newly developed prehabilitation intervention in patients undergoing hip arthroscopy for FAIS. The prehabilitation intervention was developed using a robust methodology underpinned by a systematic review and meta-analysis of the current literature and an international consensus statement. The randomised design and assessor blinding approach will minimise selection, detection, and performance biases. A well-designed prehabilitation study will positively impact patient outcomes by improving strength and function and reduce anxiety

prior to surgery thereby ensuring quicker recovery post-op. It is anticipated to reduce socio-economic burden by facilitating continuation or resumption of work and may reduce economic impact on services post-op (e.g., reduced need for physiotherapy follow-up and quicker discharge from the service). The qualitative component of this study will capture both patients and clinician perspectives of the safety, suitability and acceptability of the intervention and outcome measures and will assist in adapting the study elements prior to a full randomised control trial to test the effectiveness of prehabilitation in this population. This study is conducted within a tertiary NHS setting and therefore can be reproduced and applied in similar healthcare settings.

## Limitations of the study

There are several limitations to this study. The number of study participants has been selected to test the feasibility and to be able to perform a sample size calculation to determine the number of participants required for a definitive trial. Therefore, these results should be considered as preliminary findings. The study is conducted in a single tertiary hospital in the UK and therefore may need to be adapted prior to wider implementation in other healthcare services across the globe. As this is a feasibility study, the impact of this study on patient care and NHS costs cannot be understood fully and may not change practice. However, if found successful further funding will be sought out for a future definite trial to test its effectiveness.

## Discussion

Despite the exponential growth and development of innovative surgical interventions for FAIS, it is estimated that nearly half (49%) receiving hip arthroscopy do not achieve optimal outcomes [10]. A recent retrospective analysis of more than 4900 patients from the Non-arthroplasty Hip Registry in the UK reported that almost 35% patients undergoing hip arthroscopy for FAIS did not achieve the MCID of 13 points and only 50% achieved substantial clinical benefit (SCB) of 28 points on the iHOT-12 at 12 months post-op [8]. Deficits in muscle strength and ROM in patients with FAI is well evidenced in the literature [15, 80]. Additionally, pandemic outbreaks like COVID-19 has placed an unprecedented delay on elective operations leading to further chronicity and deconditioning in patients [81]. Therefore, optimising patients through effective prehabilitation programmes might play a vital role in early identification of impairments and promote physical and psychological wellbeing prior to undergoing surgery [82]. However, the role of exercise interventions prior to surgery in FAIS is under explored. This study will provide important preliminary data to inform feasibility of a definitive RCT in the future to evaluate effectiveness of a prehabilitation intervention. Additionally, the study will record adherence, adverse effects and satisfaction of the prehabilitation programme which are important elements at the feasibility stage [25].

It is anticipated that this programme will eventually lead to improvement in the delivery of FAIS care, reduce social and healthcare costs and assist in targeting and optimising patients much earlier in their clinical pathway.

## Supporting information

**S1 Appendix. Details of the prehabilitation intervention.**
(DOCX)

**S2 Appendix. Sample consent form.**
(DOCX)

**S3 Appendix. Objective outcome measurements.**
(DOCX)

**S4 Appendix. SPIRIT checklist.**
(DOC)

**S1 File.**
(DOCX)

## Acknowledgments

The authors would like to thank the Physiotherapy Department based at Addenbrooke's Hospital, Cambridge for their support in delivering the intervention for the study. The authors would also like to thank Physitrack PLC for permitting the use of the images.

## Author Contributions

**Conceptualization:** Anuj Punnoose, Leica Claydon-Mueller, Alison Rushton, Vikas Khanduja.

**Funding acquisition:** Anuj Punnoose.

**Methodology:** Anuj Punnoose, Leica Claydon-Mueller, Alison Rushton, Vikas Khanduja.

**Project administration:** Anuj Punnoose, Leica Claydon-Mueller.

**Resources:** Anuj Punnoose.

**Supervision:** Leica Claydon-Mueller, Alison Rushton, Vikas Khanduja.

**Validation:** Anuj Punnoose, Leica Claydon-Mueller, Alison Rushton, Vikas Khanduja.

**Visualization:** Anuj Punnoose, Leica Claydon-Mueller, Vikas Khanduja.

**Writing – original draft:** Anuj Punnoose.

**Writing – review & editing:** Anuj Punnoose, Leica Claydon-Mueller, Alison Rushton, Vikas Khanduja.

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
