## [Decision Letter · Decision Letter 0]

7 Nov 2023

PONE-D-23-28781PREHAB FAI-Prehabilitation for patients undergoing arthroscopic hip surgery for Femoroacetabular Impingement Syndrome - Protocol for an assessor blinded randomised controlled feasibility studyPLOS ONE

Dear Dr. Khanduja,

Thank you for submitting your manuscript to PLOS ONE. After careful consideration, we feel that it has merit but does not fully meet PLOS ONE’s publication criteria as it currently stands. Therefore, we invite you to submit a revised version of the manuscript that addresses the points raised during the review process.

We look forward to receiving your revised manuscript.

Kind regards,

Joshua Robert Zadro, PhD

Academic Editor

PLOS ONE

Journal Requirements:

"I have read the journal's policy and the authors of this manuscript have the following competing interests: Dr Khanduja reported being a chief investigator

for the NIHR doctoral fellowship during the conduct of the study; receiving consulting fees from Smith & Nephew

PLC and Arthrex Inc, outside the submitted work; serving as a board member for the UK Non-Arthroplasty Hip

Registry, British Hip Society, European Society of Sports Traumatology, Knee Surgery, and Arthroscopy, and the

International Society of Orthopaedic Surgery and Traumatology; and serving on the editorial boards of the Journal

of Clinical Orthopaedics and Trauma, International Journal of Orthopaedics, International Orthopaedics, and Open

Access Journal of Sports Medicine. No other disclosures were reported."

7. We note that Figure 2 in your submission contain copyrighted images. All PLOS content is published under the Creative Commons Attribution License (CC BY 4.0), which means that the manuscript, images, and Supporting Information files will be freely available online, and any third party is permitted to access, download, copy, distribute, and use these materials in any way, even commercially, with proper attribution. For more information, see our copyright guidelines: http://journals.plos.org/plosone/s/licenses-and-copyright.

9. We note that S1 File includes an image of a [patient / participant / in the study]. 

Additional Editor Comments:

This is an interesting and important protocol however there are several key issues related to the statistical analysis that need to be addressed (as raised by Reviewer #3).

Reviewers' comments:

Reviewer's Responses to Questions

**Comments to the Author**

1. Does the manuscript provide a valid rationale for the proposed study, with clearly identified and justified research questions?

Reviewer #1: Yes

Reviewer #2: Partly

Reviewer #3: Yes

2. Is the protocol technically sound and planned in a manner that will lead to a meaningful outcome and allow testing the stated hypotheses?

Reviewer #1: Yes

Reviewer #2: Partly

Reviewer #3: Partly

3. Is the methodology feasible and described in sufficient detail to allow the work to be replicable?

Reviewer #1: Yes

Reviewer #2: Yes

Reviewer #3: No

4. Have the authors described where all data underlying the findings will be made available when the study is complete?

Reviewer #1: Yes

Reviewer #2: Yes

Reviewer #3: No

5. Is the manuscript presented in an intelligible fashion and written in standard English?

Reviewer #1: Yes

Reviewer #2: No

Reviewer #3: Yes

6. Review Comments to the Author

You may also provide optional suggestions and comments to authors that they might find helpful in planning their study.

Reviewer #1: I thank the author for the opportunity to review this mixed methods study investigating the feasibility of a prehabilitation programme for patients with femoral acetabular impingement syndrome. Overall, the protocol is clear and well written. The introduction could be more specific when discussing surgical and non-surgical management. Some more specific comments are below.

Introduction

88 which outcomes showed significant improvement? And at what time point?

90 Which patient group had improvements? (Surgery vs non surgery results are not clear here when randomised control trials have shown similar function at 24 months between groups)

Participants

Will participants be given the option to have non-surgical management if they see significant improvements with prehabilitation over the 8-week period from their baseline measurements?

Reviewer #2: I would like to congratulate the efforts you have made. However, there are still some improvements could be made for this manuscript:

The introduction should be more specific, supported by evidence, and provide detailed background information to clarify the study's purpose and importance. The evidence should align closely with the research aims.

The strength of this study lies in its evidence-based approach. I suggest that the author organize the evidence more effectively to create a robust evidence statement. Ensure that the evidence provided aligns closely with the study's research aims or objectives.

I recommend that the author refine or rephrase the relevant sentences to ensure a stronger connection between the evidence and the study's purpose.

Consider allocating participants before baseline evaluation to minimize bias and enhance the validity of results.

Please find detailed comments and suggestions for improvement provided below:

Abstract:

I suggest that the author mention what outcome will be measured in the method section for better understanding.

Page 4, lines 74-76

“An epidemiological study conducted in 2015 reported a high prevalence of FAI (47%, 95% CI 42,51), within a sample (n=200, aged 16-65 years) representative of the UK population”. This sentence is a bit hard to follow, maybe try to reword the sentence and emphasize the high prevalence in the UK population with specific values, e.g. “An epidemiological study revealed a high prevalence of FAI, with 47% of the sample ….”. Please notice the correct format of the confidence interval (95%CI: [42-51]).

Page 4, lines 90-92

“However, results from these trials demonstrate that only half (51%) 91 of participants showed significant improvement on the primary outcome measure – iHOT33 (mean difference 6·8 points; 95% CI 1·7–12·0) [10]”. Does ‘these trials’ represent both surgery and physiotherapy? The author mentioned iHOT33 to demonstrate the evidence, but you need to specify what this outcome measured for, i.e., quality of life.

S1 File

The programme is based on the consensus statement as evidence-based practice, the author needs to make sure the reference is cited.

Page 9, lines 187-191

“…after prehabilitation before surgery, and at 6 weeks+/- 4 weeks and 6 months+/- 4 weeks (planned primary endpoint for definitive RCT) postoperatively when participants attend the research site for clinical care. Additionally, all patient-reported outcome measures will be captured at 12 months+/-4 weeks post-op.’’

Are the programme phases designed according to the evidence of systematic review or consensus statement? It is better to bring some evidence to illustrate why this particular timeline (in the introduction section), to strengthen the purpose of this study.

Page 9, line 201: “Chief Investigator (CI)”

The author mentioned confidence interval as CI in the previous Introduction section, but the same abbreviation appears in the Method section. Distinguish between "CI" as Chief Investigator and "CI" as a confidence interval for clarity.

Page 16 Randomization and blinding

“Following baseline evaluation, participants will be randomly allocated to one of the two groups (prehabilitation or usual care) by the PI using a computer-generated simple random allocation sequence.”

If participants are allocated after baseline evaluation, there's a risk that the group assignments could be influenced by the knowledge of individual participants' baseline characteristics, and this could introduce bias into the study. Therefore, I recommend the author consider allocating participants before any baseline evaluations or interventions applied. This approach helps maintain the validity and scientific evidence of the results. Otherwise, you need to justify why the allocation is after the baseline evaluation.

This section needs to make more efforts to improve the study’s transparency and clarity. It is better to rearrange the section's placement for better flow, i.e., put this section before the outcome measurements.

Page 24, lines 543-549

The author needs to improve the description of the study's strengths, focusing on its potential impact on future research and benefits to patients. The author should consider what the outcome could bring to future research, what benefit could bring to the patients who need surgical intervention, and whether qualitative research could have an impact on the participants.

I believe that I have been clear in my suggestions, and I am available for further clarification.

Kindest regard

Reviewer #3: This manuscript is a study protocol of a blinded randomized control trial to assess the feasibility, acceptability and safety of a FAI-prehabilitation program in patients undergoing hip surgery. The trial was registered via a valid ISRCTN number, and was approved by the respective ethics board. The study objectives are on target. However, I mostly have some concerns/comments in the statistical design and analytical framework, and CONSORT guidelines, which may require attention:

1. Randomization: More details are needed in this section, specifically, what does the authors meant by a computer-generated random allocation sequence? Furthermore, why a block randomization (often convenient and appealing in RCTs) not conducted?

2. Sample size/Power: The sample size/power writeup does not present the desired effect size that the investigators want to power the study upon; they only referr to a published paper. More details needed. Furthermore, it doesn't clearly state the specific statistical tests that was used to calculate the power/sample sizes. Moreover, given the apparent longitudinal nature of the study (statistical analysis will assess study outcome measures), sample sizes could have been computed, reflecting the actual study design (repetitions), keeping the same significance level, and same power. ANOVA could be used directly to compute the sample size, versus some simple 2/multiple group tests. Several software provide that option as well. Justification is missing in that regard.

3. Statistical Analysis: Independent t-tests are valid only under Normality assumptions. Alternative statistical methods are not discussed, under violations to the Gaussian assumptions. Furthermore, since the study will be conducted at repeated time-points, it is not clear why a GEE, a mixed model, or even an ANOVA (repeated measures, precisely) was not proposed. Authors might be only interested in evaluating the final endpoint, however, evaluating the trajectory (since data will be available at the intermediate time-points) may strengthen the analysis.

4. Writing style:

Discussion Section: Given the study was designed only for UK subjects, the Discussion section should clearly allude to future studies with other populations and geographical regions to further validate their current findings. Currently, the findings are only limited to this population.

7. PLOS authors have the option to publish the peer review history of their article (what does this mean?). If published, this will include your full peer review and any attached files.

Reviewer #1: No

Reviewer #2: No

Reviewer #3: No

---

## [Author Response · Author response to Decision Letter 0]

5 Feb 2024

The response to the editor and reviewers have been uploaded as a separate document. Please see uploaded documents

---

## [Decision Letter · Decision Letter 1]

13 Feb 2024

PONE-D-23-28781R1PREHAB FAI-Prehabilitation for patients undergoing arthroscopic hip surgery for Femoroacetabular Impingement Syndrome - Protocol for an assessor blinded randomised controlled feasibility studyPLOS ONE

Dear Dr. Khanduja,

Thank you for submitting your manuscript to PLOS ONE. After careful consideration, we feel that it has merit but does not fully meet PLOS ONE’s publication criteria as it currently stands. Therefore, we invite you to submit a revised version of the manuscript that addresses the points raised during the review process.

We look forward to receiving your revised manuscript.

Kind regards,

Joshua Robert Zadro, PhD

Academic Editor

PLOS ONE

Journal Requirements:

Additional Editor Comments:

I thank you for addressing the reviewers comments adequately. am happy to accept this manuscript once the additional comment from the statistical editor is addressed.

Reviewers' comments:

Reviewer's Responses to Questions

**Comments to the Author**

1. Does the manuscript provide a valid rationale for the proposed study, with clearly identified and justified research questions?

Reviewer #3: Yes

2. Is the protocol technically sound and planned in a manner that will lead to a meaningful outcome and allow testing the stated hypotheses?

Reviewer #3: Yes

3. Is the methodology feasible and described in sufficient detail to allow the work to be replicable?

Reviewer #3: Yes

4. Have the authors described where all data underlying the findings will be made available when the study is complete?

Reviewer #3: Yes

5. Is the manuscript presented in an intelligible fashion and written in standard English?

Reviewer #3: Yes

6. Review Comments to the Author

You may also provide optional suggestions and comments to authors that they might find helpful in planning their study.

Reviewer #3: The authors were able to address my previous comments with a great degree of satisfaction. However, I would still recommend a statement of sample size, given that feasibility trials also require a guideline wrt. sample sizes. Otherwise, I would not consider the manuscript to be full-proof.

7. PLOS authors have the option to publish the peer review history of their article (what does this mean?). If published, this will include your full peer review and any attached files.

Reviewer #3: No

---

## [Author Response · Author response to Decision Letter 1]

9 Mar 2024

All recommendations have been addressed in the attached rebuttal letter

---

## [Editor Report · Decision Letter 2]

13 Mar 2024

PREHAB FAI-Prehabilitation for patients undergoing arthroscopic hip surgery for Femoroacetabular Impingement Syndrome - Protocol for an assessor blinded randomised controlled feasibility study

PONE-D-23-28781R2

Dear Dr. Khanduja,

We’re pleased to inform you that your manuscript has been judged scientifically suitable for publication and will be formally accepted for publication once it meets all outstanding technical requirements.

Kind regards,

Joshua Robert Zadro, PhD

Academic Editor

PLOS ONE

Additional Editor Comments (optional):

I thank the authors for being so attentive to the reviewers comments. I am happy to accept this protocol and wish the authors all the best for conducting the trial.
---

## [Editor Report · Acceptance letter]

20 Mar 2024

PONE-D-23-28781R2 

PLOS ONE

Dear Dr. Khanduja, 

I'm pleased to inform you that your manuscript has been deemed suitable for publication in PLOS ONE. Congratulations! Your manuscript is now being handed over to our production team.

Kind regards, 

on behalf of

Dr. Joshua Robert Zadro 

Academic Editor

PLOS ONE